# Endometrial Receptivity–Lessons from “Omics”

**DOI:** 10.3390/biom15010106

**Published:** 2025-01-11

**Authors:** Louie Ye, Evdokia Dimitriadis

**Affiliations:** 1Reproductive Service Unit, The Royal Women’s Hospital, Melbourne, VIC 3052, Australia; 2Department of Obstetrics, Gynaecology and Newborn Health, University of Melbourne, Melbourne, VIC 3052, Australia; evdokia.dimitriadis@unimelb.edu.au

**Keywords:** endometrial receptivity, implantation, genomics, epigenomics, transcriptomics, proteomics, lipidomics, metabolomics, microbiomics

## Abstract

The window of implantation (WOI) is a critical phase of the menstrual cycle during which the endometrial lining becomes receptive and facilitates embryo implantation. Drawing on findings from various branches of “omics”, including genomics, epigenomics, transcriptomics, proteomics, lipidomics, metabolomics, and microbiomics, this narrative review aims to (1) discuss mechanistic insights on endometrial receptivity and its implication in infertility; (2) highlight advances in investigations for endometrial receptivity; and (3) discuss novel diagnostic and therapeutic strategies that may improve reproductive outcomes.

## 1. Introduction

During the menstrual cycle, the endometrium transitions from a non-receptive to a receptive state through complex molecular and cellular changes, including endometrial remodelling, decidualization of stromal cells, and the recruitment of immune cells such as uterine natural killer (uNK) cells, which collectively establish a tolerogenic environment in preparation for successful embryo implantation [1].

This highly coordinated process occurs transiently during a period of the menstrual cycle known as the “window of implantation” (WOI), typically between days 20 and 24 of the 28-day cycle (Figure 1) [2,3,4]. Endometrial receptivity (ER) refers to the state of the endometrium during the WOI where the luminal epithelium (LE) plays a critical role as the first point of contact between the maternal endometrium and the embryo [3]. Dysregulation of mechanisms controlling ER may lead to implantation failure and infertility [4,5].

A comprehensive understanding of ER is essential for development of novel diagnostic tests and therapies for infertility [4,5]. While bulk omics (e.g., genomics) have revealed key mechanisms regulating ER, single-cell omics, particularly in the area of transcriptomics, are starting to provide a more detailed, high-resolution view of molecular changes on the individual cell level that may lay the foundation for the development of personalized treatments (Figure 1) [6,7,8].

This review will highlight advances, gaps in knowledge, and future directions from various branches of omics on the endometrium, with emphasis on (1) the regulatory mechanism for ER; (2) the dysregulation of ER in infertility; and (3) the targets for clinical translation.

## 2. Lessons from Genomics

Genomic investigation in the context of ER examines the entire genome, including coding and non-coding regions of DNA. Genome sequencing of the endometrium during the WOI have provided an opportunity to discover Receptivity Associated Genes (RAGs), as well as genetic variants (e.g., single nucleotide polymorphisms—SNPs) that may affect ER and fertility outcomes.

### 2.1. Genes and Endometrial Receptivity

A comprehensive resource, the Human Gene Expression Endometrial Receptivity database (HGEx-ERdb), has catalogued 19,285 genes for their expression in the human endometrium. Within this database, 179 genes have been consistently identified as RAGs [9]. The regulation and expression of these genes and their impact on ER is discussed in a subsequent section of this review.

### 2.2. Genetic Alterations

Genetic alterations are changes in the DNA sequence or structure that can affect genes and their expression. These alterations can occur at various levels, including single nucleotides (i.e., SNPs), segments of DNA, entire genes, chromosomes, or as epigenetic modifications. SNPs have been identified in many elements of the complex network that regulate ER. In terms of cell adhesion and signalling, a polymorphism in the human Mucin 1 (*MUC1*) gene may alter its expression in endometrial epithelial cells, potentially reducing embryo implantation [10].

In terms of endometrial hormonal signalling, SNPs in the progesterone receptor (*PGR*) and estrogen receptor (*ESR1 or ESR2*) genes may lead to abnormal receptor expression, resulting in abnormal hormone receptor signalling, and ultimately leading to inadequate endometrial preparation. For instance, a polymorphism in human *PGR* (i.e., +331G/A polymorphisms) has been shown to increase the risk of implantation failure in women undergoing in vitro fertilization (IVF) [11]. In contrast, the impact of a polymorphism on an estrogen receptor in regard to ER remains to be elucidated. Whilst the polymorphism in estrogen receptor 1 (*ESR1*) may be associated with conditions characterized by abnormal ER (i.e., endometriosis), when assessed in the context of IVF outcomes for infertile women, SNPs in *ESR1* provided limited predictive value [12,13].

With regard to cell cycle regulation and apoptosis, SNPs in the human tumour protein 53 (i.e., *TP53 Arg72Pro*) may lead to dysregulation of the genes involved in apoptosis and cellular remodelling, leading to lower implantation success in women undergoing IVF [14]. A number of other SNPs in the human genome, including nuclear factor kappa beta (*NF-kβ*), leukemia inhibitory factor (*LIF*), vascular endothelial growth factor (*VEGF*), VEGF receptor 2 (*VEGFR-2*), tumour necrosis factor-alpha (*TNF-α*), interleukin-1 beta (*IL-1β*), interleukin-6 (*IL-6*), and the signal transducer and activator of transcription 3 (*STAT3*), have been linked to recurrent implantation failure (RIF), potentially causing dysregulation of angiogenesis and cytokine production [15,16,17,18,19,20,21].

## 3. Lessons from Epigenomics

Whilst alteration to gene expression can stem from changes in DNA sequences (e.g., SNPs), the field of epigenomics investigates various mechanisms regulating gene expression without altering the underlying DNA sequence. Key epigenetic mechanisms include DNA methylation and histone modification [22]. Aberrant DNA methylation and histone modification may disrupt the expression of genes essential for ER [23,24,25,26].

### 3.1. DNA Methylation and Demethylation

Stable DNA methylation requires a balance between methylation and demethylation processes. DNA methyltransferases (DNMTs) methylate DNA by adding methyl groups to DNA. In contrast, DNA demethylation is facilitated by ten-eleven translocation (TET) enzymes that remove or modify methyl groups from DNA [27]. Alternatively, passive demethylation may occur if a maintenance DNMT (i.e., DNMT1) is inactive or downregulated [28].

During early embryonic development, nearly all methyl marks in the embryonic genome are removed in a process known as global DNA demethylation that is facilitated by both active and passive demethylation [27]. Following this event, a new pattern of DNA methylation is re-established by DNMT3A and DNMT3B (de novo DNMTs) during embryo implantation [29,30]. After cell fate specification is completed during embryogenesis, the DNA methylation patterns established by DNMT3A/B are maintained by DNMT1 during subsequent cell divisions [29].

The expression of de novo DNMTs (i.e., DNMT3A/B) rather than a maintenance DNMT (i.e., DNMT1) varied across the menstrual cycle, suggesting that DNA methylation in the human endometrium is dynamic, correlating to the dynamic expression of genes vital to ER [31]. Furthermore, the DNA methylation patterns in the human endometrium may be influenced by the menstrual cycle phases where the late secretory phase and menstrual phase showed the most pronounced changes in terms of methylation profiles [32].

Genome-wide DNA methylation profiling of the human endometrium indicated that, while the overall methylome remained relatively stable during the transition from the pre-receptive to the receptive phase, 5% of the CpG sites showed differential methylation during this transition. Methylation changes affected pathways in extracellular matrix (ECM) organization, immune response, angiogenesis, and cell adhesion, highlighting the nuanced role of epigenetic regulation in preparing the endometrium for implantation [33]. Differential methylation was observed in genes associated with endometrial function and implantation, such as Transforming Growth Factor Beta 3 (*TGFB3*), Vascular Cell Adhesion Molecule 1 (*VCAM1*), and C-X-C Motif Chemokine Ligand 13 (*CXCL13*) [33].

Aberrant DNA methylation patterns during the WOI may lead to impaired ER, contributing to implantation failure and infertility. Whilst we have not identified any investigations exploring the global DNA methylation patterns in women with RIF, altered DNA methylation of ER-related genes, such as the Homeobox A10 (*HOXA10*) gene, have been reported in the eutopic endometrium of women with endometriosis [23]. Reduced endometrial epithelial and stromal *HOXA10* expression during the luteal phase have also been reported in a number of other conditions associated with infertility, such as RIF, recurrent miscarriage, and endometriosis [34,35,36].

From human and animal studies, hypermethylation of the promoter and surrounding regions of the *Hoxa10/HOXA10* may reduce its expression, thereby implicating abnormal methylation in infertility. Specific levels of DNA methylation have been reported although the range of levels is broad, limiting its utility in disease prediction. For instance, the mean methylation rate of *HOXA10* in the eutopic endometrium from women with endometriosis varied between 4 and 70% depending on the gene regions analyzed; F1-3 regions of the *HOXA10* promoter, as well as the 5′untranslated region (5′UTR) [37,38,39,40,41,42]. Hypermethylation *Hoxa10/HOXA10* have also been observed in animal models of endometriosis. In a baboon model of surgically induced endometriosis, complete and partial methylation in the F1 region of the *HOXA10* promoter was found in the eutopic endometrium [43]. Similarly, partial methylation of 10.7% in the promoter region of *Hoxa10* has been reported in the eutopic endometrium of a mouse endometriosis model. Collectively, data from human and animal models suggest that even a relatively low level of methylation in the promoter region of *Hoxa10/HOXA10* is abnormal and sufficient for disrupting normal gene expression [44].

As TET enzymes play a role in maintaining the balance of DNA methylation and demethylation during embryonic development and cellular differentiation, dysregulation of these enzymes (e.g., TET1) has been associated with the hypermethylation of DNA in various diseases [45]. A mid-secretory-phase reduction in TET1 mRNA expression in the eutopic endometrium of infertile women with endometriosis may be a potential mechanism for the hypermethylation of *HOXA10* [46]. Interestingly, mid-secretory-phase upregulation of TET3 mRNA expression in the eutopic endometrium of infertile women with endometriosis has also been reported [46]. In a recent study, using a human endometrial stromal cell (ESC) model, Liu et al. demonstrated that elevated TET3 levels in miR-29a-inhibited ESCs increased demethylation of the Collagen type 1 alpha 1 chain (Col1A1) promoter, thereby increasing the Col1A1 expression that ultimately impaired the in vitro decidualization of ESCs and reduced embryo implantation rates in a mouse model [47]. Collectively, these studies highlight the importance of TET in maintaining normal methylation balance in the context of ER. In the context of these epigenomic insights, a recent study aimed to determine whether DNA methylation patterns of genes associated with the WOI in cervical secretions can predict ongoing pregnancy outcomes in patients undergoing in vitro fertilization-embryo transfer (IVF-ET) [48]. Out of 158 genes, 15 differentially methylated probes in 14 genes were identified as important to ongoing pregnancy. These include Serpin Family E Member 1 (*SERPINE1*), Serpin Family E Member 2 (*SERPINE2*), and Transgelin-2 (*TAGLN2I*), which showed consistent methylation patterns associated with pregnancy. Validation of the three genes in an independent sample set showed consistent methylation differences between groups (i.e., ongoing pregnancy and no pregnancy). Machine learning models using the methylation data effectively classified these patterns with high predictive power (i.e., AUC ranging from 86 to 91%). A non-invasive methylation analysis of cervical secretions may serve as a diagnostic tool to predict ER and improve IVF success rates. Further validation with larger datasets is required to confirm these findings and refine the prediction models.

### 3.2. Histone Modification

Histone modification refers to a broad category of chemical changes made to histone proteins that regulate chromatin structure and gene expression. For the purpose of this review, we will focus on histone acetylation and its role in ER. Histone acetylation is a type of histone modification that involves the addition of an acetyl group to lysine residues located on the protruding tails of histones [45,49]. This type of modification is typically linked to transcriptional activation and is regulated by two opposing enzyme groups: histone acetyltransferases (HATs), which add acetyl groups, and histone deacetylases (HDACs), which remove them [49].

Histone acetylation varies during menstrual cycle. For example, a significant increase in Histone H4 acetylation at lysine 8 (H4K8ac) expression in the human endometrium was observed during the receptive phase, correlating with differentiation and preparation for implantation [50]. These acetylation changes may be under hormonal control, as demonstrated by a recent retrospective cohort study using archived endometrial samples from 40 women (20 with high progesterone levels and 20 with normal progesterone levels on the day of hCG administration). In this study, histone acetylation was assessed via immunohistochemistry, which showed an increase in Histone H3 acetylation at lysine 9 (H3K9ac) expression in the human glandular epithelium in the high-progesterone group [51].

In addition to histone acetylation expression in the endometrium, a recent study explored the regulatory mechanisms behind Histone H3 acetylated at lysine 27 (H3K27ac)-mediated gene activation during decidualization, specifically focusing on its role in regulating the Insulin-Like Growth Factor Binding Protein 1 (*IGFBP-1*) gene in human ESCs. Tamura et al. first isolated and induced decidualization in human ESCs, then performed Chromatin Immunoprecipitation Sequencing (ChIP-seq) and ChIP-qPCR to assess H3K27ac levels and transcription factor recruitment [52]. Decidualization of human ESCs induced a significant increase in H3K27ac levels at the distal promoter of the decidualization marker IGFBP-1, ultimately leading to upregulation of its expression [52]. Using a similar cell model and also performing ChIP-seq, Katoh et al. further demonstrated how upregulation of key genes (e.g., *WNT4*) in decidualization requires not only an increase in histone acetylation (e.g., H3K27c) at promoter regions of a gene but also removal of histone methylations, such as the trimethylation of lysine 27 on histone H3 (H3k27me3), to allow for transcriptional activation [53]. These findings suggest that the resolution of repressive methylation is just as important as deposition of activating acetylation for gene expression. While we have not identified any direct evidence linking H3K27ac specifically to endometriosis and RIF, the role of H3K27ac in decidualization suggests that aberrations in this histone modification could potentially contribute to implantation failures. Further research is needed to understand the role of H3K27ac in various disease phenotypes associated with infertility.

While both HDACs and histone methylation can contribute to gene silencing, HDACs primarily achieve gene repression by deacetylation, which condenses chromatin and directly blocks transcription. Studies have shown that HDAC1, HDAC2, and HDAC3 mRNA were consistently expressed in the human endometrium without significant cyclical variation [54]. However, HDAC2 protein levels exhibited a slight but significant increase during the secretory phase compared to the early proliferative phase. In contrast, HDAC1 and HDAC3 protein levels remained relatively constant throughout the cycle [54]. The observed increase in HDAC2 protein during the secretory phase may be associated with endometrial preparation for potential embryo implantation.

Aberrant histone acetylation and HDAC expression have been implicated in implantation failure and endometriosis [26]. *HDAC1/2* gene overexpression has been reported in endometriotic lesions compared to normal human endometrial tissue. HDAC1/2 proteins were expressed in both diseased and control tissues, with stronger staining being observed in endometriotic lesions [55]. The findings suggest that dysregulation of HDAC expression in endometriotic cells may lead to hypoacetylation and abnormal gene silencing, contributing to pathogenesis. A subsequent study showed that endometriotic lesions exhibited a global hypoacetylation of histone H3but not H4 compared to controls; specific lysine residues (H3K9 and H4K16) were significantly hypoacetylated in endometriotic lesions [24]. Furthermore, Monteiro et al. observed hypoacetylation at the promoter regions of genes (e.g., *HOXA10*) previously also implicated in ER [24].

Despite the theoretical potential, there are currently no clinical trials or approved treatments involving HAT modulators for infertility [56]. The complexity of epigenetic regulation and the challenges in developing specific and safe HAT modulators have limited progress in this area. HDAC inhibitors (HDACis) such as trichostatin A (TSA) have been reported to enhance decidualization in human endometrial stromal cells by increasing histone acetylation and upregulating decidualization markers [57,58]. There are no registered clinical trials investigating HDACis specifically for the treatment of female infertility. Preclinical research suggests the potential therapeutic applications of HDACis in endometriosis; however, randomized clinical trials (RCTs) are essential to confirm the safety and efficacy before clinical implementation [59].

Other forms of histone modifications that have been implicated in abnormal ER and are potentially amenable to modulation include histone ubiquitination and lactylation. For instance, inhibition of the monoubiquitination of histone H2A (H2AK119ub1) disrupted decidualization and resulted in pregnancy failure in a mouse model [60]. Similarly, a reduction in lactylation of H3K18 (H3K18la) was associated with pregnancy failure in a sheep model [61]. Collectively, these studies highlight the complex roles histone modifications play in orchestrating ER and the need for future investigations to better understand their implication in human infertility.

## 4. Lessons from Transcriptomics

A transcriptomic investigation of ER examines the complete set of RNA transcripts, including messenger RNA (mRNA) and non-coding RNA (ncRNA) expressed during the WOI. While the mRNA expression ER-related genes have been identified and extensively studied, post-transcriptional regulation of mRNA has become an area of intense research in recent years. ncRNA such as microRNA (miRNA) and long non-coding RNA (lncRNA) have been shown to play a key role in post-transcriptional regulation of the expression of genes involved in a variety of cellular process essential for ER [62,63].

### 4.1. messengerRNA

Of the 179 RAGS identified from the HGEx-ERdb, 151 were consistently upregulated during the receptive phase [9]. Since this earlier study, we have found that the number of genes implicated in ER varies across studies, reflecting the complexity of this biological process. Transcriptomic analyses have identified between 107 and 2878 genes that are differentially expressed when comparing pre-receptive and receptive phases of the human endometrium [64].

These genomic insights have paved the way for advanced diagnostic tools and personalized strategies to improve implantation success. One such innovation is Endometrial Receptivity Analysis (ERA), a genomic diagnostic test that evaluates the expression of 238 genes to determine the ER status. By pinpointing the ideal timing for embryo transfer (ET), ERA may enhance the success rates of assisted reproductive technologies (ART), offering hope to individuals facing implantation challenges [65].

Despite the promise of ERA, current evidence is insufficient for supporting its routine clinical application [66]. More recently, a systematic review was performed which included studies that investigated ERA-guided euploid embryo transfer cycles focusing on live birth rate (LBR), ongoing pregnancy rate (OPR), implantation rate (IR), clinical pregnancy rate (CPR), biochemical pregnancy loss rate (BPLR), and miscarriage rate (MR). The review included 11 studies with 7581 patients, of which 1663 underwent ERA. The authors found that ERA did not significantly improve reproductive outcomes, such as LBR, OPR, or CPR, in either the general infertile population or patients with a history of RIF [67].

### 4.2. microRNA

miRNAs are small, non-coding RNA molecules, typically 20–24 nucleotides long, that regulate gene expression at the post-transcriptional level. They function by binding to complementary sequences in the 3′ untranslated regions (3′-UTRs) of mRNAs, leading to either mRNA degradation or translational repression. A number of miRNA and their respective targets, as well as putative roles in ER, have been identified in recent years. Table 1 provides some key examples.

Earlier studies demonstrated that miR-30b and miR-30d are upregulated in the receptive endometrium whilst miR-494 and miR-923 are downregulated in the receptive endometrium [76]. By comparing miRNA target predictions with previous mRNA microarray data, 12 genes were identified as potentially significant for ER, including Calpastatin (*CAST*), Cystic Fibrosis Transmembrane Conductance Regulator (*CFTR*), Fibroblast Growth Factor Receptor 2 (*FGR2*), and *LIF* [76]. A more recent meta-analysis of transcriptomic studies has validated 19 miRNAs (e.g., members from the miR-30, and miR-200) with 11 corresponding upregulated meta-signature genes involved in ER [77]. From animal model and human cell line experiments, miRNAs have been shown to regulate RAGs by different mechanisms (e.g., overexpression) across different cell types (e.g., endometrial stromal cells) involved in ER.

A cellular transition process that is critical to ER is mesenchymal–epithelial transition (MET), where stromal cells adopt an epithelial phenotype characterized by improved adhesion and polarity. During decidualization, maternal endometrial stromal cells undergo a MET to support embryo implantation. Using both mouse and human stromal cell models, Jimenez et al. showed that, during decidualization of endometrial stromal cells, the miR-200 family upregulates epithelial markers and downregulates Zinc finger E-box-binding homeobox 1/2 (*ZEB1/2*), promoting MET. Conversely, inhibition of miR-200 maintains *ZEB1/2* expression, preventing MET [78]. Similarly, using both the mouse in vivo model and a human epithelial cell model, upregulation of miR-494-3p was shown to impair ER by targeting LIF and modulating the PI3K/AKT/mTOR pathway. [79]. Furthermore, the regulation of ER is not confined to the endometrium alone. Vilella et al. demonstrated that Hsa-miR-30d was significantly upregulated during the WOI compared to other menstrual phases [80]. miR-30d was secreted by endometrial epithelial cells primarily as an exosome-associated molecule. Mouse embryos internalized both free and exosome-associated miR-30d through the trophectoderm, which led an increased expression of genes involved in cell adhesion (e.g., Integrin subunit beta 3). Embryos treated with hsa-miR-30d exhibited a significant increase in adhesion to the endometrial epithelial cell monolayer compared to control embryos [80].

Aberrant miRNA expression has been linked to a number of conditions associated with endometrium-related infertility, including RIF, endometriosis, and polycystic ovarian syndrome (PCOS) [23,51,81,82,83,84,85,86]. For instance, downregulation of miR-9 and miR-34 families in the eutopic endometrium of women with endometriosis may delay progesterone-induced shifts to secretory phase endometrium by maintaining the persistent expression of anti-apoptotic (e.g., BCL2) and proliferation genes (e.g., Cyclin E2), thereby impairing ER [85]. Similarly, downregulation of other miRNA families has been implicated in the dysregulation of ER. A recent study analyzing endometrial samples from RIF patients showed that the downregulation of miR-30d-5p corresponded to an elevated level of the suppressor of cytokine signalling 1 (SOCS1—an inhibitor of LIF) expression and reduced the expression of LIF and p-STAT3 (markers critical for ER and implantation) [87].

While no miRNA therapy is currently available, miRNA-based classifiers using miRNA expression profiles from 200 IVF patients have been created to identify WOIs to assist in identifying the optimal time for ET in IVF cycles. The classifier is based on 21 miRNAs that were differentially expressed across three time points after progesterone administration (e.g., 108 ± 5 h). These miRNAs were linked to critical pathways in ER and embryo implantation. The classifier demonstrated high accuracy (93.9% training set, 88.5% testing set) in identifying the optimal WOI for ET [88]. miRNAs are stable and robust biomarkers that may serve as alternatives to traditional mRNA-based receptivity assays (e.g., ERA). Using the miRNA-based classifier for WOI determination may improve IVF outcomes. This approach has the potential to be expanded to study other endometrium-related infertility conditions such as RIF and endometriosis.

### 4.3. Long Non-Coding RNA

lncRNAs are transcripts longer than 200 nucleotides that do not code for proteins but regulate gene expression at various levels [89]. Unlike miRNA, lncRNAs do not exclusively regulate gene expression at the post-transcriptional level, and they are capable of also regulating gene expression at the pre-transcriptional, transcriptional, and post-translational stages. One well-known mechanism of action of lcnRNA relevant to ER regulation is its ability to act as a molecular sponge, sequestering miRNAs, thereby preventing the inhibition of target mRNA expression [90].

For example, lncRNA H19 can inhibit the activity of miRNA let-7 by acting as a sponge/decoy or competitive endogenous RNA (ceRNA) (Figure 2A–C). This inhibition prevents let-7 from downregulating its target genes, such as integrin β3 (*ITGB3*), thereby reducing adhesion and the invasive ability of trophoblastic cells [91]. While previous studies have shown that downregulation of H19 lncRNA and *ITGB3* in the human endometrium is associated with unexplained pregnancy loss and RIF, the exact role of H19/let-7/ITGB3 axis in unexplained pregnancy and RIF remains to be determined [92,93].

While lncRNA can inhibit miRNA expression, reciprocal inhibition has also been reported (Figure 2D). Overexpression of miR-200c suppressed the expression of lncRNA MALAT1 in endometrioid endometrial cancers, subsequently downregulating the expressions of mesenchymal proteins (e.g., Vimentin) essential for epithelial–mesenchymal transition (EMT), a critical feature of tumour progression and metastasis [94]. While dysregulation of this MALAT1-miR200c axis has been shown in endometrial cancer, its impact on ER and embryo implantation remains to be determined.

While the mechanisms by which lncRNA regulates miRNA is gaining clarity, the factors that regulate lncRNA in endometrial epithelial cells remain elusive. Takamura et al. exposed human endometrial epithelial cells (HEECs) to blastocyst-conditioned media (BCM) from embryos that either implanted or failed to implant after IVF to investigate how a lncRNA—Phosphatase and Tensin Homolog Pseudogene 1 (PTENP1) regulates HEECs’ adhesive capacity [95]. The authors found that PTENP1 was upregulated in HEECs exposed to BCM from successfully implanted embryos, by contrast silencing PTENP1 significantly decreased HEEC adhesion to trophoblast spheroids [95]. The study highlighted that PTENP1 may be an important regulator of ER and that embryos may modulate ER through secreted factors that fine-tune cellular and molecular pathways.

More recently, Huang et al. explored the molecular mechanisms of RIF by constructing a ceRNA network to identify potential hub genes linked to poor ER. Analysis of the GSE111974 dataset revealed 1500 upregulated mRNAs and three lncRNAs, and 1022 downregulated mRNAs and four lncRNAs in RIF samples compared to controls. The ceRNA network pinpointed five hub genes, such as Gap Junction Alpha-1 (*GJA1*). While the hub genes themselves are mRNAs, the constructed lncRNA-miRNA-mRNA ceRNA networks indicate that specific lncRNAs interact with these hub genes indirectly through competitive binding with shared miRNAs. These lncRNAs may regulate the expression of the hub genes and influence the pathways involved in RIF [96]. Key hub lncRNA that were differentially expressed in the endometrium of women with RIF compared to control include prostate androgen-regulated transcript 1 (PART1), microRNA-17 Host Gene (MIR17HG), H19, and Long Intergenic Non-Protein Coding RNA 173 (LINC00173) which may play a role in cell adhesion and motility [96]. Lin et al. identified eight other lncRNAs that may be unique to women with RIF, highlighting their involvement in ER [97]. Furthermore, three candidate drugs (miconazole, terfenadine, and STOCK1N-35215) were proposed for targeting the ceRNA networks associated with RIF [97].

Nuclear Paraspeckle Assembly Transcript 1 (NEAT1) is another lncRNA that has been implicated in RIF. NEAT1 binds to CCCTC-binding factor (CTCF) to suppress HOXA10 expression via histone modification, which may impair endometrial epithelial cell proliferation and receptivity [98]. In contrast, downregulation of NEAT1 may enhance ER [98]. Several other lncRNA important in endometrial stromal cell decidualization have been implicated in RIF in recent years, including Long Intergenic Non-Protein Coding RNA 2190 (LINC02190) [99], HOXA11 Antisense RNA (HOXA11-AS) [100], and Lung Cancer-Associated Transcript 1 (LUCAT1) [101]. Furthermore, Long Intergenic Non-Protein Coding RNA 1960–201 (LINC01960–201) is another lncRNA that has been implicated in the decidualization of endometrial stromal cells in women with endometriosis-associated infertility during the WOI. Abnormal expression of LINC01960-201 may disrupt the LINC01960-201/ADAMTS7/miR-608 axis, impairing ER and potentially leading to recurrent miscarriage. [102]. Collectively, these findings highlight the complex regulatory roles of lncRNAs in ER and their potential as therapeutic targets.

### 4.4. Single-Cell and Spatial Transcriptomics

Single-cell and spatial transcriptomics have provided us with new tools to investigate the molecular mechanisms regulating ER. By analyzing gene expression at the individual cell level, researchers have identified molecular maps of the endometrium during the WOI. Two recent studies have utilized single-cell RNA sequencing (scRNA-seq) to compare endometrial samples from women with RIF and those from fertile women [7,8]. Both studies have shown that scRNA-seq is superior to traditional bulk RNA sequencing because it allows for the analysis of cell-specific expression (e.g., disruptions to estrogen and progesterone signalling in epithelial cells), cell–cell interactions (e.g., WNT5A-SFRP4 signalling between stromal and epithelial cells), gene expression in rare subpopulations of cells (e.g., natural killer cell subsets such as CD49a + CXR4+ cells), and gene expression within a specific cell population over time [7,8]. Furthermore, a recent study combined scRNA-seq with a spatial transcriptomics approach to characterize the human endometrium [103]. This combined approach allowed the analysis of cell specific gene expression to be mapped to defined spatial regions within the endometrium. Overall, these studies demonstrated novel approaches to explore cell behaviour in complex and heterogenous tissue, like the endometrium, that traditional bulk methods could not achieve.

## 5. Lessons from Proteomics

Proteomics analysis involves studying the entire protein complement of a cell, tissue, or organism, including protein expression, structure, interactions, and modifications. The most commonly used method in proteomics is mass spectrometry (MS), often coupled with liquid chromatography (LC-MS/MS) [104]. This combination is considered the gold standard in proteomics due to its versatility, sensitivity, and ability to provide comprehensive insights into protein identification, quantification, and characterization.

Proteomic analysis can be performed on endometrial biopsies and endometrial fluid aspirate (EFA). The first study on endometrial fluid protein content was conducted by Beier and Beier-Hellwig in 1998 [105]. In recent years, EFA has become more common due to the distinct advantage it has over endometrial biopsies, as it is associated with less discomfort and risk for patients and better represents the local uterine environment leading up to implantation. However, a significant limitation to using EFA is the lack of standardization in collection methods and timing. Standardization may help to reduce variability in sample quality and biomolecule concentration in order to improve the reliability of proteomic analyses and facilitate the identification of potential biomarkers for fertility-related disorders.

By analyzing endometrial tissue, Chen et al. identified 196 proteins differentially expressed from the endometrium during the mid-proliferative (MP) versus mid-secretory (MS) phase [106]. Differentially expressed proteins identified were linked to biological processes essential for ER, such as cell structure, motility, immunity, and developmental processes. Specific pathways, including c-Jun N-terminal kinase (Jnk) signalling and Epidermal Growth Factor (EGF) signalling, were enriched during the MS phase, highlighting their role in preparing the endometrium for implantation [106]. Examples of validated MS-phase endometrial epithelial proteins that may play a role in ER and implantation include Rho GDP-dissociation inhibitor alpha (Rho-GDIα) and Chloride Intracellular Channel Protein 1 (CLIC1) [106].

Similarly, proteins were differentially expressed in EFA during MP compared to the MS phase of the menstrual cycle. More importantly, EFA proteins were differentially expressed between fertile and infertile women, highlighting that EFA proteins are linked to receptivity and fertility. Key endometrial epithelial proteins, such as antithrombin III and alpha-2-macroglobulin, were implicated in implantation [107]. Another study profiled the endometrial secretome using EFA collected during the luteal phase. The authors identified 82 proteins differentially expressed between prereceptive (LH+4) and receptive (LH+9) phases [108]. Proteins upregulated in the receptive phase (LH+9) were primarily involved in host defence and immune responses, while downregulated proteins were linked to stress response and cell structure. Fourteen proteins also showed evidence of altered post-translational modifications, highlighting the complexity of proteomic regulation during the WOI [108].

More recently, Kasvandik et al. compared the proteomic landscape of EFA obtained from fertile women and women with RIF. The authors found that, in women with RIF, 21 proteins showed expression levels similar to the early secretory phase in fertile controls, indicating a displacement of the WOI [109]. Furthermore, a 4-protein panel consisting of Progesterone Reception (PGR), Nicotinamide N-Methyltransferase (NNMT), Solute Carrier Family 26 Member 2 (SLC26A2), and Lipocalin 2 (LCN2) demonstrated high specificity and sensitivity (91.7% and 96.6%) for distinguishing between receptive and non-receptive endometrium, a model that, as such, may help identify the receptive endometrium and optimize the timing for ET in IVF cycles [109].

A number of other approaches have been developed to evaluate the proteome of the receptive endometrium. One recent study identified 82 proteins relevant to the implantation process from extracellular vesicles (EVs) secreted by human endometrial epithelial cells. Key proteins linked to the process of implantation include Annexins (ANXA2, ANXA4, ANXA5), Integrins (ITGA1, ITGA2, ITGAV, ITGB1), and Mucins (MUC1, MUC4, MUC16) [110]. Another approach utilized isobaric tags for relative and absolute quantitation (iTRAQ—a multiplex approach used in quantitative proteomics) to analyze pre-receptive (LH+2) and receptive (LH+7) endometrial tissue [111]. The authors identified 173 differentially expressed proteins with integrated analyses, highlighting five hub proteins, the most significant being Acyl-CoA Synthetase Long Chain Family Member 4 (ACSL4), which was increased in the receptive phase and reduced in patients with RIF [111]. Knockdown of ACSL4 reduced the expression of key ER markers such as HOXA10, LIF, and Cyclooxygenase (COX) and impaired the adhesive capacity of epithelial cells in vitro [111].

Using a variety of the aforementioned methodologies, such as iTRAQ and LC-MS/MS, a number of studies investigating receptive phase endometrial tissue and fluid have consistently identified differential protein expression between fertile women and women with RIF. Notably, Annexin-6 (ANXA6), PGR, and metalloproteinase-2 and 9 were differentially expressed in the endometrium of women with RIF [112]. Proteins such as Mucin 16 (MUC16, also known as CA125), glycogen phosphorylase B (PYGB), and Tubulin Polymerization-Promoting Protein Family Member 3 (TPPP3) were identified as potential biomarkers for assessing ER and predicting IVF outcomes [112]. Collectively, these proteomic insights are crucial for developing diagnostic tools and therapeutic strategies aimed at improving implantation success rates.

## 6. Lessons from Lipidomics

Lipidomics is the comprehensive study of lipids within biological systems. Lipid mediators secreted by the endometrium include triglycerides and eicosanoids such as prostaglandins, thromboxanes, leukotrienes, endocannabinoids, and sphingolipids [113,114,115,116]. Among these, endocannabinoids, lysophosphatidic acid (LPA), and prostaglandins (PG) have been investigated extensively.

Two key endocannabinoids involved in implantation are anandamide (N-arachidonoyl ethanolamine, AEA) and 2-arachidonoylglycerol. Abnormal levels of these lipids are associated with delayed implantation and poor pregnancy outcomes in a rodent model [117]. In another animal study, high uterine fatty acid amide hydrolase (FAAH) expression reduced AEA levels, creating a favourable environment for embryo implantation [118]. Conversely, low FAAH activity results in elevated AEA concentrations, which can hinder implantation [118]. By comparing plasma anandamide (AEA) levels and components of endocannabinoid system (ECS) between viable and non-viable first-trimester pregnancies in asymptomatic women presenting for early pregnancy ultrasound, plasma AEA levels were significantly higher in non-viable pregnancies compared to viable pregnancies. However, non-viable pregnancies also showed increased FAAH expression in trophoblasts and decidua in contrast to the low FAAH activity seen in rodent model, possibly as a compensatory response to elevated AEA [119]. A number of differences may account for the discrepancy between the studies, including localized versus systemic effects, sampling timing (i.e., pre-implantation in animal model), differences in species, and the measuring of FAAH activity versus expression. These differences highlight the complexity of endocannabinoid regulation in early pregnancy and suggest that AEA-FAAH dynamics may operate differently in viable vs. non-viable pregnancies. Future studies should aim to measure AEA levels, FAAH expression, and enzymatic activity at both local (uterine) and systemic levels to reconcile these findings.

Lysophosphatidic acid (LPA) is a water-soluble phospholipid and a signalling molecule with diverse functions across organs including regulation of FAAH expression and activity [120]. In earlier mice studies, LPA3 signalling was shown to regulate prostaglandin biosynthesis via cyclooxygenase-2 (COX-2) to ensure proper implantation timing and embryo spacing [114]. Subsequent animal studies also demonstrated that LPA enhanced COX-2 expression and prostaglandin E2 (PGE2) production, as well as increased FAAH expression and activity, thereby reducing endocannabinoid levels (i.e., AEA) which are harmful at high concentrations during implantation [118]. LPA also increased markers of decidualization (e.g., Insulin-Like Growth Factor Binding Protein 1) and vascularization (interleukin-10), highlighting its role in endometrial preparation [120]. LPA3 mRNA and protein levels were significantly higher in the early and late secretory phases compared to the proliferative and mid-secretory phases. LPA3 protein was localized in the cytoplasm of luminal and glandular epithelial cells during the early secretory phase. Elevated LPA3 expression in the early secretory phase may contribute to endometrial preparation by influencing COX-2, MMPs, and collagen production [121].

By comparing the expression of enzymes and signalling molecules involved in prostaglandin synthesis in endometrial biopsies obtained during the WOI from patients with RIF undergoing IVF versus fertile controls, Achache et al. found evidence of impairment in the prostaglandin synthesis pathway (e.g., significant reduction in COX-2) and LPA3 downregulation in the secretory endometrium [122]. In contrast, increased levels of PGE2 and PGF2α were identified in the EFA collected during the WOI [123].

More recent studies have identified lipid species and metabolic pathways associated with successful implantation as well as biomarkers to predict ER. For instance, Matorras et al. profiled the entire lipidome from EFA collected from women undergoing IVF to identify lipids that were important in implantation [124]. Eight lipid species were significantly altered in cycles without implantation, including seven glycerophospholipids (e.g., lysophosphatidylethanolamines—LPE and phosphatidylcholines—PC) and one omega-6 polyunsaturated fatty acid (docosapentaenoic acid). Cycles without implantations showed lower levels of certain lipids, such as LPE (20:5), while others like PC (40:8) were higher. Altered lipid profiles likely reflect changes in metabolic pathways related to inflammation, membrane composition, and nutrient availability in the endometrium. A support vector machine (SVM) algorithm using the eight significant lipid metabolites achieved an AUC of 89.3% [124].

In contrast, another study evaluated lipid ion ratios in EFA as non-invasive biomarkers for assessing ER and distinguishing between receptive (positive pregnancy outcome) and non-receptive (implantation failure) cycles [125]. The authors identified 13 lipid ion ratios were significantly elevated in the non-receptive group and identified phosphatidylethanolamines, diacylglycerols, and sphingolipids as key contributors to non-receptivity. The findings suggest that steroid metabolism and LDL remodelling play a role in ER. Furthermore, Braga et al. demonstrated that lipid ion ratios can be used as an non-invasive prediction model in freeze-all cycles with an AUC of 84% [125]. Collectively, by analyzing lipid signatures, clinicians may better determine the WOI for ET in IVF cycles to improve implantation outcomes.

## 7. Lessons from Metabolomics

Metabolomics is the comprehensive analysis of metabolites within biological systems. Whilst some evidence suggests that the metabolomics of ECM could predict embryo viability and implantation rates better than traditional morphology-based assessments, a recent systematic analysis found that RCTs using a metabolomics approach did not show a significant impact on key outcomes such as CPR or LBR [126]. Currently, metabolomics is not supported for clinical use in terms of improving fertility outcomes. However, by examining the metabolic profiles from endometrial tissues or serum samples, researchers are beginning to identify specific metabolites and pathways that may be important in ER and implantation.

In a recent study, an untargeted metabolomic profiling of endometrial samples was conducted using ultra-high-performance liquid chromatography-tandem mass spectroscopy (UPLC-MS/MS) to identify associations between metabolomes and causes of infertility [127]. Endometrial samples were collected from women diagnosed with infertility, categorized based on etiology such as endometriosis, RIF, unexplained infertility, or male factor infertility. A total of 925 metabolites were identified, with lipids being the most abundant, especially polyunsaturated fatty acids (PUFAs) like linoleate and linolenate [127]. Women with endometriosis or RIF exhibited lower levels of PUFAs compared to those with male factor or unexplained infertility. The study highlighted the potential role of PUFAs in ER and contribution it may have in the pathogenesis of endometriosis and RIF.

In contrast, Zheng et al. aimed to discover metabolites involved in endometrial transformation and RIF from serum samples [128]. Serum was collected from women undergoing hormonal replacement therapy (HRT)-frozen embryo transfer (FET) cycles on the day of progesterone administration and on the third day of progesterone administration. Controls include 19 RIF patients and 19 controls. Gas chromatography–mass spectrometry (GC-MS) was utilized for the metabolomic analysis. A total of 105 serum metabolites were identified initially, with significant reductions in 76 during the initial 3 days of endometrial transformation, which was associated with reduced amino acid metabolism and suppression of the tricarboxylic acid (TCA) cycle. The reduced metabolic activity was associated with lower serum levels of amino acids (e.g., glutamic acid, ornithine, proline) during endometrial transformation, particularly in patients with RIF [128]. In terms of potential clinical applications, eight metabolites (e.g., indol-3-propionic acid) were highly predictive of RIF, with a strong discriminative ability between controls and RIF patients (i.e., AUC > 70% for these metabolites) [128]. These findings suggest that serum metabolite profiling could serve as a non-invasive method for assessing ER and predicting implantation outcomes in FET cycles. Furthermore, biomarkers like malic acid and indol-3-propionic acid could guide personalized interventions to optimize implantation success.

Whilst specific interventions are not currently available to modulate the endometrial metabolome, dietary interventions (i.e., Mediterranean diet) have been suggested [127]. A diet rich in Omega-3 fatty acids (e.g., fish, flaxseeds, and walnuts) may support a favourable endometrial environment since Omega-3 and Omega-6 (PUFAs) play a role in PG production, which is important in ER. Excessive Omega-6 intakes may disrupt the balance necessary for proper endometrial transformation and implantation by promoting pro-inflammatory responses [129]. In a study by Molina et al., the authors reported that a high adherence to the Mediterranean Diet positively correlated with beneficial metabolites like progestin steroids and ceramides and negatively correlated with potentially harmful metabolites like bile acids and acylcarnitines in women with endometrial-factor infertility [127]. Linolenate levels were notably higher in women with high Mediterranean Diet adherence and no endometrial-factor infertility compared to women with low Mediterranean Diet adherence and endometrial-factor infertility [127]. Adopting a Mediterranean Diet could positively modulate the endometrial metabolomic profile, potentially improving fertility outcomes.

## 8. Lessons from Microbiomics

The endometrial microbiome is the community of microorganisms residing in the endometrial lining. Over the past decade, research has increasingly focused on the relationship between the endometrial microbiome and ER, particularly exploring how the microbial environment of the endometrium influences fertility outcomes.

A healthy endometrial microbiome is typically dominated by Lactobacillus species. Studies have shown that a Lactobacillus-dominant microbiome (LDM) was associated with higher implantation and pregnancy rates for women undergoing IVF [130,131]. Non-Lactobacillus-dominant microbiome (NLDM) with potentially pathogenic species such as Gardnerella, Streptococcus, Atopobium, Burkholderia, and Prevotella was associated with poorer reproductive outcomes from IVF cycles [130,132].

Similarly, an increased presence of NLDM was associated with RIF and recurrent miscarriage [133,134]. Verstraelean et al. identified a Bacteroides and Proteobacteria dominance in endometrial brush samples from 19 non-pregnant women with a history of RIF or recurrent miscarriage [133]. A subsequent analysis of paired EFA and vaginal secretion samples collected during the luteal phase from 28 RIF patients and 18 infertile women linked the presence of Burkholderia with RIF [134].

Collectively, assessing for LDM may serve as a biomarker for predicting implantation success. Furthermore, assessing and potentially modifying a NLDM (e.g., probiotic or targeted antibiotic) may present as a potential therapeutic options to improve reproductive outcomes. Clinical trials have explored the utility of antibiotics and probiotics to address implantation failure by modulating the endometrial microbiome.

In one pilot study, patients with NLDM received antibiotics followed by prebiotics and probiotics to restore Lactobacillus dominance. Whilst the intervention successfully restored the LDM in a small number of patients, the reproductive outcomes were not as expected, as untreated patients showed higher pregnancy rates than the treated group, highlighting the need to consider better risk stratification (i.e., by severity of dysbiosis) in order to identify subgroups of patients who would benefit the most from treatment [132].

In a prospective pilot study, 117 infertile women with RIF and 55 infertile women without RIF (control group) were treated with oral enteric-coated lactoferrin to restore a LDM with the aim of improving reproductive outcomes. Lactoferrin supplementation changed NLDM to LDM in 43.2% of women with RIF. Furthermore, an improved LDM was associated with higher CPR and LBR [135]. Whilst the study highlighted the potential of microbiome-targeted therapies, RCTs are needed to confirm these results and optimize lactoferrin dosage and duration.

In addition to evaluation of novel therapies, a number of fundamental biological questions remain to be answered, particularly regarding the mechanism by which an altered endometrial microbiome negatively impacts ER and implantation. The prevailing theory suggests that the influence an endometrial microbiome has on ER and implantation is due to its interaction with local immune cells [136,137]. For instance, a healthy microbiome may promote immune tolerance by enhancing the activity of regulatory T-cells (Tregs) and reducing pro-inflammatory responses [137].

In a recent study, He et al. revealed several ways by which bacterial populations may improve ER and reproductive outcomes [138]. Using an animal model of chronic endometritis (CE) first, treatment with Lactobacillus crispatus significantly reduced CE-induced endometrial inflammation and improved uterine morphology. Implantation rates were restored to near-normal levels in the combined treatment group (i.e., antibiotics and probiotics) [138]. When the combined treatment (antibiotics + L. crispatus vaginal administration during IVF) was given to 100 infertile women with CE, the authors of the study observed higher CPR than with antibiotics only or control groups [138]. Furthermore, a histological analysis showed reduced inflammatory cell infiltration and improved endometrial structural morphology in samples from women who received combined treatment [138]. In addition to influence on the local immune environment, L. crispatus upregulated ER-related proteins such as progesterone, VEGF, and MMP9, highlighting the complexity of the microbiome’s influence on ER.

Whilst microbiomics continue to provide new insights into ER, significant challenges in research methodology remain to be addressed: (1) variability in sample collection, analysis methods, and patient selection across studies has hindered the comparability of results; (2) standardization is critical to defining a healthy endometrial microbiome and its clinical implications [139]. The authors proposed a standardized pipeline to (1) focus on consistent sampling during the same menstrual cycle phase; (2) using advanced sequencing methods (e.g., next-generation sequencing, metagenomics) targeting specific 16S rRNA regions (V3-V4); (3) employing controls to prevent contamination and ensure reproducibility [139].

## 9. Lessons from Integratomics

Single-omics studies (e.g., transcriptomics) often provide a limited view of a system because they focus on one layer of regulation. The term “integratomics” refers to the comprehensive integration of various ’omics’ data in order to achieve a holistic understanding of complex biological systems. This integrative approach is increasingly common in contemporary research and is gradually becoming the standard approach to unravelling the complex cellular processes involved in ER.

While the concept of a multi-omics approach is no longer novel, its application in specific contexts provides valuable insights and has demonstrated the method’s power to address complex biological questions. As an example, a recent study by Tang et al. integrated transcriptomics, proteomics, and functional assays to investigate the role of Cell Division Cycle 42 (CDC42) deficiency in RIF [140]. Whilst transcriptomic analysis identified the pathways and genes impacted by CDC42 loss, proteomics and functional assays validated these findings at the protein and cellular function levels. Integrated analyses uncovered the interplay between CDC42, Wnt signalling, and decidualization, providing a mechanistic understanding of CDC42’s role in RIF [140].

Another recent study also exemplified the multi-omic approach as it integrated multiple data layers, including morphology, transcriptomics (single-cell and spatial), proteomics, and protein–protein interaction (PPI) analysis to comprehensively investigate the cellular and molecular mechanisms underlying decidualization resistance (DR) [141]. Each omic level validated and extended findings from the other, enhancing reliability and offering deeper mechanistic insights, showing that disrupted responses to progesterone and oestrogens, along with cytoskeletal and oxidative stress imbalances, are the main drivers for DR in the context of severe preeclampsia [141]. These findings have implications beyond preeclampsia, extending to conditions like endometriosis and RIF, where DR has also been implicated.

Collectively, the aforementioned examples underscore the utility of an integrative multi-omics approach in elucidating the complex biological underpinnings of ER and conditions that impact ER, paving the way for more accurate diagnostic tools and personalized therapeutic interventions.

## 10. Conclusions

In summary, various disciplines of the omics have revolutionized our understanding of ER, uncovering key targets for advancing diagnostic and therapeutic strategies in reproductive health.

Genomic studies linked SNPs in genes such as *PGR* and *TP53* to poorer IVF outcomes, while SNPs in *NF-kβ*, *LIF*, *VEGF, VEGFR-2, TNF-α, IL-1β, IL-6*, and *STAT3* are implicated in RIF. Epigenomic studies revealed a range of potential targets that may be amenable to modulation, including DNMTs, TETs, HDAC1/2, H3K9, H4K16, H2AK119ub1, and H3K18la. A transcriptomic analysis revealed key miRNAs and lnCRNA implicated in endometriosis and RIF, including miR-9, miR-34, and miR-135a/b miR-30d-5p, PART1, MIR17HG, H19, LINC00173, NEAT1, and LINC01960-201.

Proteins such as PGR, MMP2/9, and ANXA6 may be targets for further research due to their association with RIF. Lipid targets, including endocannabinoids, prostaglandins, LPA, and PUFAs offer additional avenues to improve reproductive outcomes. Finally, a Lactobacillus-dominant microbiome emerges as a critical element for the establishment of a receptive environment for implantation.

Collectively, these insights provide a robust foundation for developing personalized diagnostic and therapeutic strategies to enhance implantation success and improve reproductive outcomes. Future research should aim to integrate multi-omic datasets to achieve a comprehensive understanding of the complex biological processes underpinning ER.

## Figures and Tables

**Figure 1 biomolecules-15-00106-f001:**
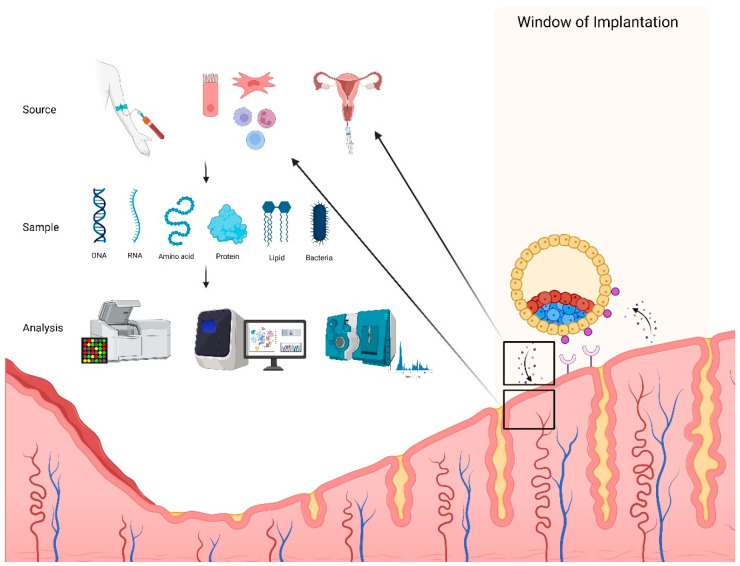
Omics analysis of endometrial receptivity. A myriad of biomolecules have been extracted from a variety of sources, including serum, endometrial cells, and endometrial fluid aspirate, during the window of implantation for complex analysis. This figure was created in https://BioRender.com (accessed on 13 December 2024).

**Figure 2 biomolecules-15-00106-f002:**
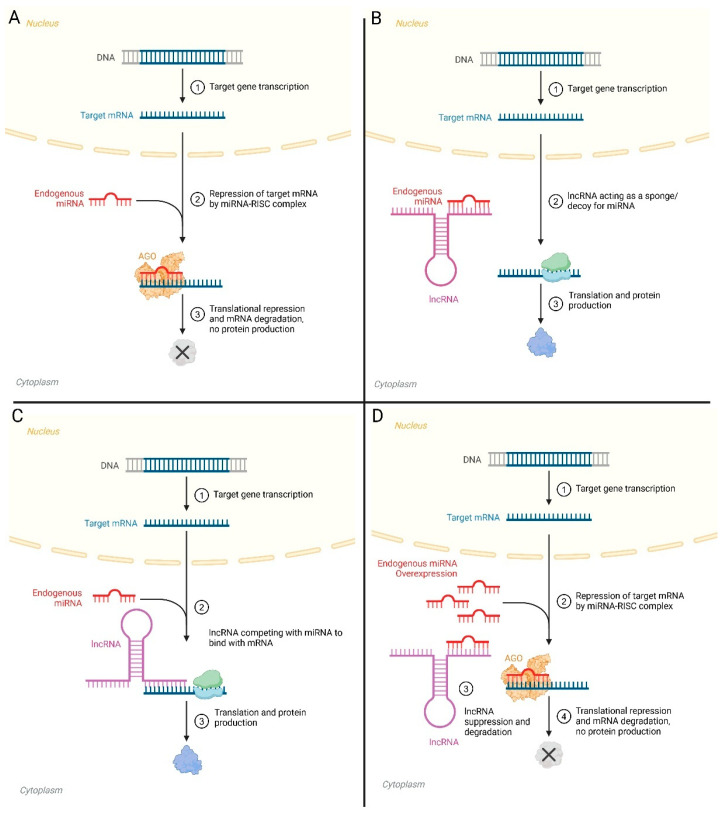
Interaction between lncRNA, miRNA and mRNA. (**A**) miRNA inhibiting mRNA expression; (**B**) lncRNA inhibiting miRNA to promote mRNA expression; (**C**) lncRNA competes with miRNA to bind to mRNA, promoting its expression; (**D**) miRNA inhibiting mRNA and lncRNA expression. Abbreviations—AGO = Argonaute proteins, miRNA-RISC complex = microRNA-induced silencing complex. This figure was created in https://BioRender.com (accessed on 26 December 2024).

**Table 1 biomolecules-15-00106-t001:** miRNAs and endometrial receptivity from human and animal models.

miRNA	Targets ^1^	Function/Disease Association	Species	Reference
miR-145	IGF1R	Embryo attachment	Human	[68]
miR-140	IGF1R	Embryo attachment	Rat	[69]
miR-27a-3p	IGF1	Chronic endometritis	Human	[70]
miR-223-3p	LIF	Endometrial receptivity	Mouse	[71]
miR-181	LIF	Embryo attachment	Mouse	[72]
miR-126a-3p	Itga11	Embryo attachment	Mouse	[73]
miR-135a/b	HOXA10	Endometriosis	Human	[74]
miR-124-3p	IQGAP1	Embryo attachment	Mouse/Human	[75]

Abbreviations—IGF1R = Insulin-like Growth Factor 1 Receptor, IGF1 = Insulin-like Growth Factor 1, Itga11 = Integrin Subunit Alpha 11, IQGAP1 = IQ Motif Containing GTPase Activating Protein 1, ^1^ endometrial target.

## Data Availability

No new data were created or analyzed in this study. Data sharing is not applicable to this article.

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
