# Peer review of "Endometrial Receptivity–Lessons from “Omics”"

_biomolecules, 2025, doi:10.3390/biom15010106_

Round 1

Reviewer 1 Report

Comments and Suggestions for Authors

In this work, the authors comprehensively review the results of omics analyses on endometrial receptivity during the implantation window. The subject of the review is novel and relevant to the field. The writing is clear and easy to follow, and the topic is well-presented and referenced. The authors begin each section with an introductory paragraph that explains the concept under discussion. These paragraphs are particularly valuable for readers who may not be familiar with the specific mechanisms involved.

The review summarizes the contributions of various omics approaches (from genomics to microbiomics). However, for sections 2 (genomics) and 3 (epigenomics), the authors would strengthen the review by specifying the species (human, mouse, etc.) covered by the cited studies. This information is consistently provided throughout the remainder of the article.

The article includes a figure and a table to illustrate key concepts. To further enhance understanding, the authors could consider incorporating a figure in section 4.2 that visually depicts the complexity of lncRNA/miRNA/mRNA interactions. 

Author Response

Comment 1. The review summarizes the contributions of various omics approaches (from genomics to microbiomics). However, for sections 2 (genomics) and 3 (epigenomics), the authors would strengthen the review by specifying the species (human, mouse, etc.) covered by the cited studies. This information is consistently provided throughout the remainder of the article.

Response 1: Thank you for pointing this out. We agree with this comment. Therefore, we have now specified species in both Section 2 and 3. These changes can be found throughout Section 2 and 3 in the following locations – line 85, 90, 98, 101, 244, 252. Furthermore, new content added to Section 3.1 from line 155-185, Section 3.2 from line 221-240, 271-278 also now specifies species covered by the cited studies.

Comment 2. The article includes a figure and a table to illustrate key concepts. To further enhance understanding, the authors could consider incorporating a figure in section 4.2 that visually depicts the complexity of lncRNA/miRNA/mRNA interactions. 

Response 2: Thank you for making this suggestion. We agree with you that a visual representation will help readers to better understand the complexity of lncRNAs/miRNA/mRNA interactions. Therefore, we have added a new Figure 2 to illustrate lncRNA/miRNA/mRNA interactions. Figure 2 can now be found under section 4.3 between Line 440-441.

Reviewer 2 Report

Comments and Suggestions for Authors

"Endometrial Receptivity – Lessons from ‘Omics’" is a comprehensive, and well-written review paper exploring various -omics approaches in the context of endometrial receptivity. The paper primarily aims to describe the impact of different fields on the regulatory mechanisms of endometrial receptivity, its dysregulation, and potential clinical translation targets. It effectively covers recent studies on diverse aspects of endometrial receptivity regulation. However, several areas could be enhanced to improve the article's overall quality and depth:

1.    The introduction (Lines 17–37) lacks sufficient references to support its claims. Adding appropriate citations would strengthen this section.

2.    The paragraph on genomics (Paragraph 2) includes information related to transcriptomics (Lines 46–54), which would be more appropriately placed in paragraph 4.

3.    The discussion of DNA methylation (Paragraph 3.1) should distinguish between changes in de novo methylation (DNMT3A/3B) and maintenance methylation (DNMT1) DNMTs, as well as include the role of demethylation processes (e.g., TET enzymes). Additionally, it would be valuable to elaborate on the specific levels of DNA methylation changes required to induce observable phenotypic changes, as well as the variability in methylation across different genomic regions.

4.    The section on histone acetylation (Paragraph 3.2) should be expanded to include studies on other histone modifications beyond H4K8ac and H3K9ac. Furthermore, the inclusion of more specific studies utilizing chromatin immunoprecipitation (ChIP) techniques would provide deeper insights into the molecular mechanisms.

5.    The paper would benefit from a discussion on the interplay between different -omics approaches within the context of endometrial receptivity. For instance, exploring correlations between epigenomics and transcriptomics or the impact of microbiomics on proteomics would provide a more integrated perspective.

6.    The conclusion (Paragraph 9) should be refined to specify the main potential targets identified through various omics approaches for improving endometrial receptivity.

Author Response

Comment 1: The introduction (Lines 17–37) lacks sufficient references to support its claims. Adding appropriate citations would strengthen this section.

Response 1: Thank you for pointing this out. We agree with this comment. Therefore, we have added 8 references to the introduction section to further support the claims. Furthermore, minor changes were made to 1) of days of menstrual cycle that defines WOI which have been changed from '19 and 24' to '20 and 23' on line 24 2) repositioned “Figure 1” to the end of the last sentence of the 3rd paragraph on line 34.  

Comment 2: The paragraph on genomics (Paragraph 2) includes information related to transcriptomics (Lines 46–54), which would be more appropriately placed in paragraph 4.

Response 2: Thank you for the suggestion. We agree with this comment. Therefore, we have moved information related to transcriptomics to Section 4. As a result, we have re-written the introductory paragraph for Section 2b between line 44-48. We have deleted part of Section 2.1 between line 51-52 and moved/rewritten line 57-77 to a new Section 4.1 between line 290-310.

Comment 3: The discussion of DNA methylation (Paragraph 3.1) should distinguish between changes in de novo methylation (DNMT3A/3B) and maintenance methylation (DNMT1) DNMTs, as well as include the role of demethylation processes (e.g., TET enzymes). Additionally, it would be valuable to elaborate on the specific levels of DNA methylation changes required to induce observable phenotypic changes, as well as the variability in methylation across different genomic regions.

Response 3: Thank you for this recommendation. We agree with this comment. Therefore, we have included a detailed discussion of DNMT3a/3b, TET enzymes and the level of DNA methylation required to induce observable phenotypic changes and variability in methylation across different genomic region under Section 3.1. These changes can be found between line 117-129, 130, 155-185.

Comment 4: The section on histone acetylation (Paragraph 3.2) should be expanded to include studies on other histone modifications beyond H4K8ac and H3K9ac. Furthermore, the inclusion of more specific studies utilizing chromatin immunoprecipitation (ChIP) techniques would provide deeper insights into the molecular mechanisms.

Response 4: Thank you for this recommendation. We agree with this comment. Therefore, we have included other examples of histone modification such as acetylation, ubiquitination, and lactylation and referenced studies using ChIP techniques. The changes can be found under Section 3.2 between line 221-240, and 271-278.

Comment 5: The paper would benefit from a discussion on the interplay between different -omics approaches within the context of endometrial receptivity. For instance, exploring correlations between epigenomics and transcriptomics or the impact of microbiomics on proteomics would provide a more integrated perspective.

Response 5: Thank you for this suggestion. We agree with this comment. Therefore, we have created a new section - Section 9 to show the latest examples of multiomic integration. These changes can be found between line 731-759.

Comment 6: The conclusion (Paragraph 9) should be refined to specify the main potential targets identified through various omics approaches for improving endometrial receptivity.

Response 6: Thank for you this recommendation. We agree with this comment. Therefore, we have re-written the conclusion specifying main potential targets identified through various omics approaches. These changes can be found between line 762-781.

Round 2

Reviewer 2 Report

Comments and Suggestions for Authors

The authors took into account the recommendations, and the revised and supplemented manuscript is suitable for publication.